# Genetic diversity, phylogenetic and phylogeographic analysis of *Anopheles culicifacies* species complex using ITS2 and *COI* sequences

**R. A. S. Rathnayake**[1], **W. M. M. Wedage**[2], **L. S. Muthukumarana**[1], **B. G. D. N. K. De Silva**[1,2,3]*

**1** Department of Zoology, Faculty of Applied Sciences, University of Sri Jayewardenepura, Nugegoda, Sri Lanka, **2** Center for Biotechnology, Department of Zoology, Faculty of Applied Sciences, University of Sri Jayewardenepura, Nugegoda, Sri Lanka, **3** Genetics and Molecular Biology Unit, Faculty of Applied Sciences, University of Sri Jayewardenepura, Nugegoda, Sri Lanka

* nissanka@sci.sjp.ac.lk

**Data Availability Statement:** All relevant data are within the manuscript and its Supporting Information files.

## Abstract

*Anopheles culicifacies* is the major vector of malaria in Sri Lanka and the Indian subcontinent which is characterized as a species complex with five sibling species provisionally designated as A, B, C, D and E. The current study was carried out to understand the phylogenetic and phylogeographic relationships between the sibling species of the species complex while observing their genetic diversity and genetic differentiation. Thirty-five ITS2 and seventy-seven *COI* sequences of *An. culicifacies* species complex reported from different geographical locations of Asia and China at the NCBI public database were used for the analysis. Bayesian likelihood trees were generated for the phylogenetic analysis. The divergence of the species complex was obtained from the Bayesian phylogeographic model in BEAST. There were two clades of the sibling species of *An. culicifacies* species complex as A, D and B, C and E in both phylogenetic and phylogeographic analysis using ITS2 sequences. Based on the highly divergent *COI* sequences and the high mutation rate of the mitochondrial genome, there were four and three clades in both phylogenetic and phylogeographic analysis using *COI* sequences. The diversification of *An. culicifacies* species complex was obtained as ranging from 20.25 to 24.12 Mya and 22.37 to 26.22 Mya based on ITS2 and *COI* phylogeographic analysis respectively. There was a recent diversification of the sibling species A and D than the sibling species B, C and E. Low haplotype diversity was observed in the sequences reported from Sri Lanka in both ITS2 and *COI* analysis that can be due to bottlenecks resulting from the intense malaria control efforts. A high genetic differentiation was achieved for some populations due to the large geographical distance. The high genetic diversity based on the five sibling species implies the possibility of maintaining a relatively high effective population size despite the vector control efforts.

**Funding:** The author(s) received no specific funding for this work.

**Competing interests:** The authors have declared that no competing interests exist.

## Introduction

*Anopheles culicifacies* Giles 1901 is the major vector of malaria in Sri Lanka [1, 2]. It belongs to the order Diptera, sub-order Nematocera, family Culicidae, genus *Anopheles*, subgenus Cellia and series Myzomyia [3]. *Anopheles culicifacies* has a wide distribution that extends from Ethiopia, Iran, Yemen westward to Cambodia, Vietnam, Laos, China eastward whereas the Indian subcontinent towards the northern part, Sri Lanka, and southern part of Thailand towards the southern part [4, 5]. In 1977, *An. culicifacies sensu lato* (s.l.) was colonized initially by the National Institute of Malaria Research (previously Malaria Research Centre) in the laboratory [3]. It has been characterized as a species complex with five sibling species provisionally designated as A, B, C, D and E [2, 3] by using cytogenetic methods. The sibling species of the complex are morphologically similar and reproductively isolated. The only sibling species B and E are reported from Sri Lanka while country where all the members of the species complex are found is in India [2, 6].

According to the polytene chromosome diagnostic banding pattern similarities between the sibling species, A and D are identified as closely related to each other than the sibling species B, C and E. However, the sibling species B, C and E are known to be closely related to each other [7]. The sequence differences of 28S-D3 rDNA domains and ITS2 rDNA region also supported the classification of the species complex into two groups A, D and B, C, E [4, 8, 9]. Vectorial capacity, differences in biting and resting behavior, resistance to insecticides, and susceptibility to malaria parasites of each sibling species vary [10]. Species A, B, C and D are mainly zoophilic and show low anthropophilic index (1–4%) than species E (80%) [11]. However, species A and E are considered as main vectors in the Indian subcontinent than species C and D whereas species B is considered as poor or non-vector [12]. It has been recorded so far that only sibling species B and E are present in Sri Lanka where species E plays the role as major vector [6].

Based on the limited sequence data available at that time it was proposed that the sibling species A and B evolved from a common ancestor. The sibling species D is considered as diverged from sibling species A and the sibling species C from sibling species B in a later period. The origin and distribution of sibling species E persist unclear [9].

The evolutionary potential and history of species are reflected from the genetic diversity and phylogenetic diversity. A higher genetic diversity implies a higher evolutionary potential and a vast ability to respond to environmental changes [13]. In depth understanding of how species evolve through genetic changes can be obtained through phylogenetic analysis and it provides the pathway that connects extant species with their ancestral origin, as well as allows us to predict the genetic divergence that may occur in the future. The phylogeography can be considered as the conceptual bridge connecting processes acting on populations with evolutionary patterns detectable among species or higher taxa [14]. These studies allow us to understand the geographic ordination of genotypes and help to understand both biogeography and landscape genetics across a variety of spatial and temporal scales [15].

The mitochondrial *COI* and *COII*, D3 (28S rDNA), ITS1 and ITS2 of ribosomal DNA (rDNA) play a significant role in species identification and phylogenetic analyses. ITS2 region is popular in studying phylogenetics of closely related *Anopheles* species, as well as biodiversity and geographic races of a particular species of mosquitoes as it is a non-coding DNA sequence containing high degree of mutations [16]. ITS2 region is used in distinguishing members of several *Anopheles* species complexes including *Anopheles maculatus* group [17]. *Anopheles hyrcanus* group [18], *Anopheles dirus* complex [17]. The sequences of both ITS2 and *COI* were used in previous studies of *Anopheles* species also. The barcode region of the five prime (5') end of the *COI*, three prime (3') end of the *COI* and ITS2 region of rDNA were used to study

about the hypothesis of new molecular lineages within *Anopheles punctimacula* s.l. [19]. The ITS2 and *COI* sequences were utilized in detection of *Anopheles* species such as *Anopheles arabiensis* and *Anopheles pretoriensis* from understudied regions of eastern Ethiopia [20].

The *COI* gene followed by ITS2 is the most commonly used molecular markers or barcode regions for mosquito barcoding studies. Comparison of rDNA ITS2 and mtDNA *COI* efficacies for the correct classification of the species has revealed that ITS2 is preferred on the *COI* gene [21].

Hence aim of this study was to maximum utilization of the data repository in the public domain, National Center for Biotechnological Information (NCBI) to infer the phylogenetic and phylogeographic relationships of *Anopheles culicifacies* species complex using ITS2 and *COI* sequences.

## Materials and methods

### Data collection

The sequences of ITS2 and *COI* of *An. culicifacies* species complex available at NCBI public database (www.ncbi.nlm.nih.gov) were used (S1 and S2 Tables). Four sequences of *Aedes aegypti* and two sequences of *Anopheles subpictus* were used as the outgroup of the phylogenetic and phylogeographic analyses respectively (S3 Table). A total number of thirty- five sequences of ITS2 and seventy- seven *COI* sequences were used for the analysis.

The number of ITS2 and *COI* sequences from each sibling species is shown in Table 1 and the number of ITS2 and *COI* sequences from each country is shown in Table 2.

### Sequence alignment

All the downloaded sequences of ITS2 and *COI* were aligned using ClustalW in MEGA5.2.2 software separately [22]. Both ITS2 and *COI* alignment files were trimmed separately to obtain a consistent region as the length of the sequences are different [23]. The final length of ITS2 aligned sequences was 439bp and *COI* aligned sequences were 450bp.

### Phylogenetic analysis

The FASTA format trimmed files of ITS2 and *COI* after aligning were converted to Nexus alignment (.nex) and Phylip alignment (.phy) file formats by Geneious Prime 2022.1.1 software. The best schemes of models of evolution for ITS2 and *COI* sequences were obtained using PartitionFinderV1.1.1_Windows software as shown in Table 3. Phylogenetic trees of ITS2 and *COI* were generated separately using obtained best models of evolution in MrBayes-V 3.2.5_ Bayesian inference of Phylogeny software.

**Table 1. Number of ITS2 and *COI* sequences from each sibling species of *An. culicifacies*.**

| Sibling Species | Number of ITS2 sequences | Percentage (%) | Number of *COI* sequences | Percentage (%) |
|---|---|---|---|---|
| A | 8 | 22.86 | 12 | 15.58 |
| B | 10 | 28.57 | 21 | 27.27 |
| C | 3 | 8.57 | 4 | 5.19 |
| D | 4 | 11.43 | 1 | 1.30 |
| E | 2 | 5.71 | 3 | 3.90 |
| Unidentified sequences up to sibling species level | 8 | 22.86 | 36 | 46.75 |
| Total sequences | 35 | | 77 | |

**Table 2. Number of ITS2 and *COI* sequences of *An. culicifacies* from each country.**

| Country | Number of ITS2 sequences | Percentage (%) | Number of *COI* sequences | Percentage (%) |
|---|---|---|---|---|
| India | 21 | 60.00 | 38 | 49.35 |
| Iran | 6 | 17.14 | 3 | 3.90 |
| Sri Lanka | 3 | 8.57 | 25 | 32.47 |
| Cambodia | 3 | 8.57 | 1 | 1.30 |
| China | 2 | 5.71 | - | - |
| Pakistan | - | - | 8 | 10.40 |
| Oman | - | - | 1 | 1.30 |
| Arab | - | - | 1 | 1.30 |
| Total sequences | 35 | | 77 | |

## Haplotype network construction using ITS2 and *COI* sequences

DNA Sequences Polymorphism software (DnaSP) (Version 5.10.01) and Network 10.2 software were used to determine the spatial distribution of haplotypes of ITS2 and *COI* sequences of *An. culicifacies* using the median-joining method. Arlequin version 3.5.2.2 software was used to obtain the haplotype frequencies of each country [24].

## Determination of genetic diversity

Genetic diversity of ITS2 and *COI* sequences *An. culicifacies* species complex was obtained for the total number of sites (excluding sites with gaps / missing data) [25]. The number of variable sites, GC content, the total number of mutations, number of haplotypes, haplotype diversity, nucleotide diversity and neutrality tests of Tajima's D, Fu and Li's D* test statistic, Fu and Li's F* test statistic, Fu's Fs statistic were obtained for both ITS2 sequences and *COI* sequences using DnaSP v5 software.

## Determination of genetic differentiation

Genetic differentiation of ITS2 and *COI* sequences *An. culicifacies* species complex among populations of the study was estimated based on fixation index ($F_{ST}$) values [26] using Arlequin version 3.5.2.2 software.

## Phylogeographic analysis

Phylogeographic analysis using ITS2 and *COI* sequences following the Bayesian phylogeographic model was carried out in BEAST v1.8.2 software [27, 28]. The diversification date of the most recent common ancestor (MRCA) of the *Anopheles* genus at 83.23 Mya with a 95% credibility interval ranging from 54.33 to 115.88 Mya [29] was used to calibrate the node ages

**Table 3. Best models of evolution obtain for ITS2 and *COI* sequences.**

| Region/ Gene | Fragment length | Coding positions | Best model of evolution |
|---|---|---|---|
| **ITS2** | 439bp | 1–439 | K80 |
| | | 2–439 | JC |
| | | 3–439 | JC |
| ***COI*** | 450bp | 1–450 | HKY + G |
| | | 2–450 | HKY + G |
| | | 3–450 | HKY + G |

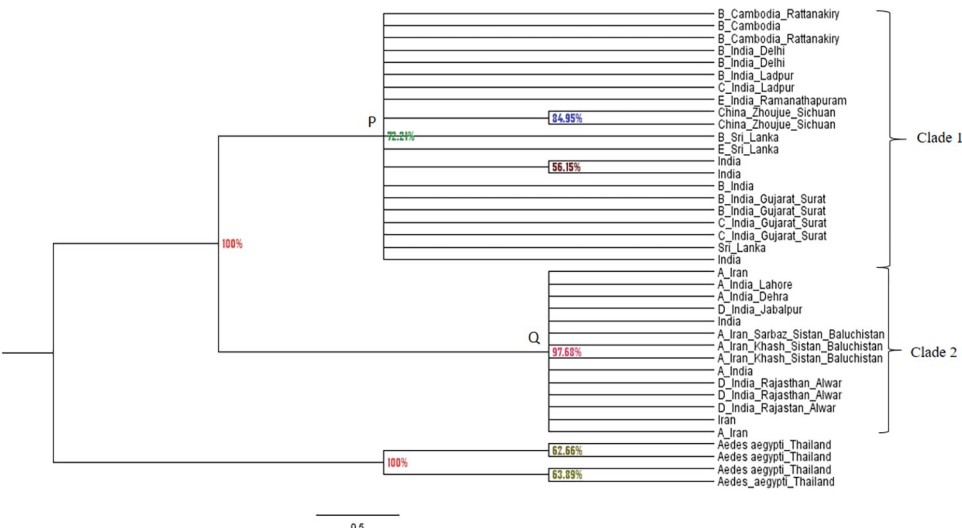

**Fig 1. Bayesian likelihood tree generated by MrBayes-3.2.5_WIN32_x86 software using ITS2 sequences of *An. culicifacies*.** 35 sequences, 439 characters, 460 000 generations, 2 mcmc runs. Percentages on the node labels are Bayesian posterior probabilities (PP). *Ae. aegypti* was used as the out-group. GenBank accession numbers of ITS2 sequences included in clade 1, 2 and outgroup were listed in S4 Table.

which was dated using the *Drosophila-Anopheles* divergence at 260 Mya suggested by Gaunt and Miles [30].

## Results

### Phylogenetic analysis of *An. culicifacies* species complex using ITS2 sequences

The generated ITS2 Bayesian likelihood tree was dispersed into two main distinct clades and each clade evolved separately having a different genetic structure for each (Fig 1). The sibling species B, C and E were included in clade 1 and the sibling species A and D were included in clade 2. The sibling species of clade 1 were reported from geographical areas of India, Sri Lanka and Cambodia and China. The sibling species of clade 2 were reported from India and Iran.

The branch length of clade 2 was greater than the branch length of clade 1 implying that sibling species of A and D of clade 2 might have more divergence than the sibling species of B, C and E in clade 1 from their most recent common ancestor (MRCA). All sibling species of A, B, C, D and E were reported in India. Only sibling species B and E were reported in Sri Lanka and sibling species A was reported from Iran and the B was reported from Cambodia. According to the Fig 1, the posterior probability value in the node of clade 1 was 72.21% (node A). The posterior probability value in the node of clade 2 was 97.68% (node B). As the clade 2 posterior probability value is close to 100 than the clade 1 value, the branching pattern of clade 2 can be considered as statistically more reliable than clade 1.

### Phylogenetic analysis of *An. culicifacies* species complex using *COI* sequences

A different topology is shown in the tree produced from the phylogenetic analyses of *COI* sequences (Fig 2) than in the ITS2 Bayesian likelihood tree. The Bayesian likelihood tree of *COI* sequences was broadly dispersed into two in the first branching point with a posterior

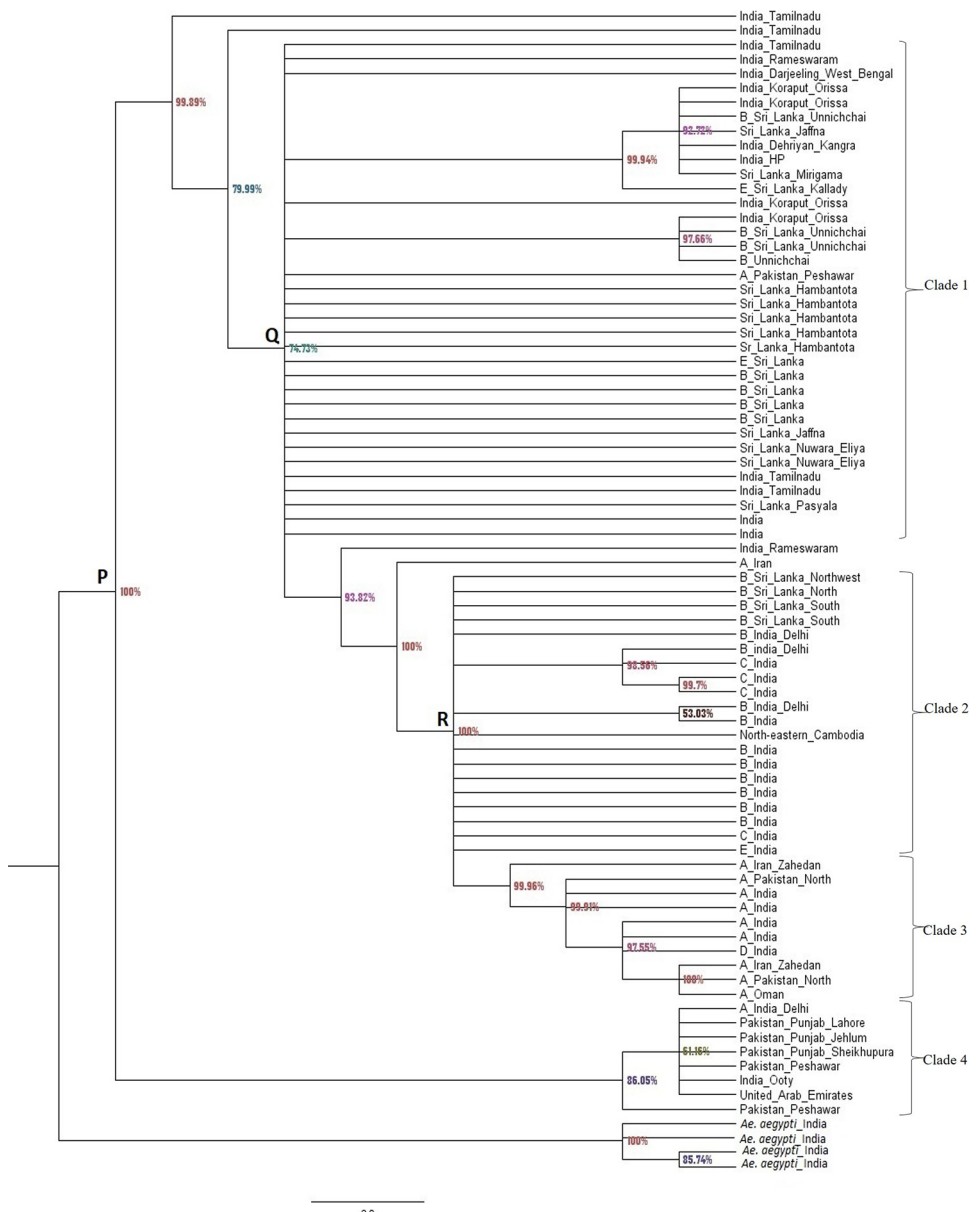

**Fig 2. Bayesian likelihood tree generated by MrBayes-3.2.5_WIN32_x86 software using *COI* sequences of *An. culicifacies*.** 77 sequences, 450 characters, 1 640 000 generations, 2 mcmc runs. Percentages on the node labels are Bayesian posterior probabilities (PP). *Ae. aegypti* was used as the out-group. GenBank accession numbers of *COI* sequences included in Clade 1 to 4 and outgroup were listed in S5 Table.

probability value of 100% in the branching point (node A). Four clades can be distinguished and each clade was separately evolved having a different genetic structure for each. The clade 1 to 3 were more closely related to each other than clade 4 with a MRCA representing node B with a posterior probability value of 74.73%.

When considering clades 1, 2 and 3, there was a more close relationship between clades 2 and 3 having MRCA representing node C with a posterior probability value of 100% implying that the branching pattern was also more statistically reliable. The clade 4 was identified as a

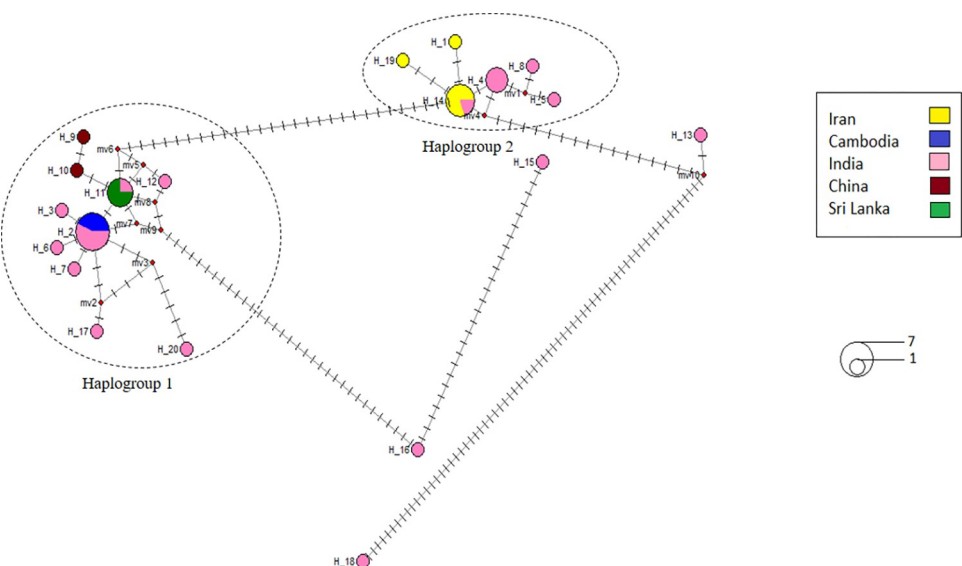

**Fig 3. ITS2 haplotype network of *An. culicifacies* species complex (20 haplotypes) generated using DnaSP v5 software and Network 10.2 software.** The size of a circle indicates the relative frequency of sequences belonging to a certain sequence type. Each color indicates a different geographic area. The distance between two haplotypes includes no. of mutations. Red colored dots indicate median vectors.

derived clade. There was a sister relationship formed by the clade 4 with other clades with its position supported by a high posterior probability value of 86.05%.

The sibling species B, C and E were included in clade 2 from India (sibling species B, C, E), Sri Lanka (sibling species B) and North-eastern Cambodia. The sibling species A and D were included in clade 3 where A from Iran, India, Pakistan and Oman and D from India. The sibling species from India, Pakistan and the United Arab Emirates were included in clade 4.

According to the Fig 3, the branch length of clade 3 was greater than the branch length of clade 2 implying the sibling species of A and D of clade 3 might have more genetic change (or divergence) than the sibling species of B, C and E in clade 2 from their MRCA. That result was the same with ITS2 phylogenetic analysis where the sibling species A and D might have more genetic divergence than sibling species B, C and E from their MRCA.

## ITS2 haplotype network of *An. culicifacies* species complex

The ITS2 haplotype network is shown in Fig 3. There was a total of 20 haplotypes recorded from Iran, Cambodia, India, China and Sri Lanka based on the 35 ITS2 sequences used in the analysis. The haplotype frequencies and haplotype diversities (Hd) in each population are listed in S6 Table. There were 3 shared haplotypes named as H_2, H_11 and H_14 and the other remaining haplotypes were private haplotypes.

The highest number of private haplotypes (13) were recorded from India (Hd = 0.729) as the number of sequences is also high in India among other countries. There were no private haplotypes from Cambodia and Sri Lanka each having one shared haplotype with India. Hd values were zero in Cambodia, China and Sri Lanka. Any of the haplotypes from China was not shared between other localities.

There were two main haplogroups. The haplogroup 1 was composed of haplotypes from India, Cambodia, Sri Lanka and China. There were predominantly private haplotypes from India and two shared haplotypes (H_2 and H_11). The regional ancestor of the haplogroup 1

**Table 4. Sibling species belong to each haplogroup of ITS2 haplotype network and reported countries.**

| Haplogroup | Sibling species | Country |
|---|---|---|
| **Haplogroup 1** | A | Iran |
| | D | India |
| **Haplogroup 2** | B, C, E | India |
| | B | Cambodia |
| | B, E | Sri Lanka |
| | Sequences not identified up to sibling species level | China |

was H_2 which is a shared haplotype between India and Cambodia that was understood as the genetic backup of the haplogroup 1.

The haplogroup 2 was made up of private haplotypes from Iran and India with one shared haplotype H_14 that is shared between those two populations. The regional ancestor of the haplogroup 2 was H_14 which is a shared haplotype between Iran and India.

The highest distance was shown between the H_18 and H_13 haplotypes of India having the highest no. of mutations between them. The Table 4 below shows the sibling species belonging to each haplogroup and reported countries.

## *COI* haplotype network of *An. culicifacies* species complex

The *COI* haplotype network is shown in Fig 4. There was a total of 55 haplotypes recorded from Iran, Sri Lanka, Pakistan, Oman, India, Cambodia and the United Arab Emirates based on the 77 *COI* sequences used in the analysis.

The haplotype frequencies and haplotype diversities (Hd) in each population were listed in S7 Table. There were 4 shared haplotypes named as Hap_4, Hap_13, Hap_16 and Hap_32 and the remaining ones were private haplotypes.

The highest number of private haplotypes (26) were recorded from India (Hd = 0.980) while the least number of private haplotypes (one from each) were recorded from Oman,

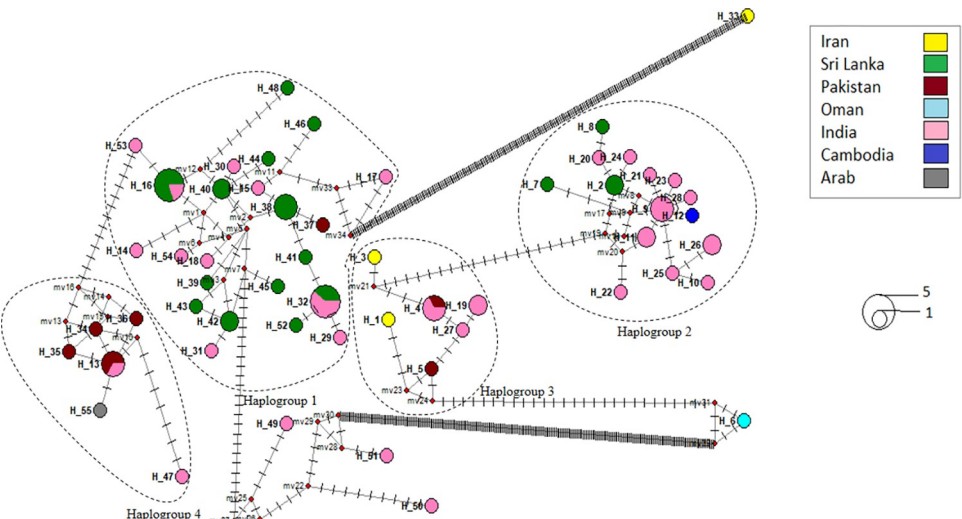

**Fig 4. *COI* haplotype network of *An. culicifacies* species complex (55 haplotypes) generated using DnaSP v5 software and Network 10.2 software.** The size of a circle indicates the relative frequency of sequences belonging to a certain sequence type. Each color indicates a different geographic area. The distance between two haplotypes includes no. of mutations. Red colored dots indicate median vectors.

**Table 5. Sibling species belongs to each haplogroup of COI haplotype network and reported countries.**

| Haplogroup | Sibling species | Country |
|---|---|---|
| **Haplogroup 1** | B, E | Sri Lanka |
| | A | Pakistan |
| | Sequences not identified up to sibling species level | India |
| **Haplogroup 2** | B, C, E | India |
| | B | Sri Lanka |
| | Sequences not identified up to sibling species level | Cambodia |
| **Haplogroup 3** | A,D | India |
| | A | Pakistan |
| | A | Iran |
| **Haplogroup 4** | A | India |
| | Sequences not identified up to sibling species level | Pakistan, Arab |

United Arab Emirates and Cambodia. The lowest Hd value was obtained from Sri Lanka (Hd = 0.957). The haplotypes of United Arab Emirates, Iran, Cambodia and Oman were not shared between other localities.

There were four main haplogroups. The haplogroup 1 was made up of haplotypes from India, Sri Lanka and Pakistan. There were predominantly private haplotypes from Sri Lanka and two shared haplotypes (Hap_16, Hap_32) that were shared between India and Sri Lanka. The regional ancestor of the haplogroup 1 was identified as Hap_40 from India.

The haplogroup 2 was made up of haplotypes from India, Sri Lanka and Cambodia. There were predominantly private haplotypes from India and no shared haplotypes. The regional ancestor of the haplogroup 2 was identified as Hap_9 of India.

The haplogroup 3 was made up of haplotypes from India, Pakistan and Iran. There was one shared haplotype Hap_4 that was shared between India and Pakistan. The regional ancestor of the haplogroup 3 was identified as Hap_4 which is a shared haplotype between Pakistan and India.

The haplogroup 4 was composed of haplotypes from India, Pakistan and the United Arab Emirates. There were predominantly private haplotypes from Pakistan and one shared haplotype Hap_13 that was shared between India and Pakistan. The regional ancestor of the haplogroup 4 was identified as Hap_13 which is a shared haplotype between Pakistan and India. The Table 5 below shows the sibling species belongs to each haplogroup and reported countries.

## Determination of genetic diversity

The genetic diversity of *An. culicifacies* species complex based on ITS2 and *COI* sequences is shown in Table 6 obtained using DnaSP Version 5.10.01 software.

When considering all sequences of ITS2 and *COI*, the nucleotide diversity ($\pi$), haplotype diversity (Hd), and an average number of nucleotide differences (k) were higher in *COI* sequences than in ITS2 sequences.

**Table 6. Number of variable sites (S), haplotypes (h), nucleotide diversity ($\pi$), haplotype diversity (Hd), the average number of nucleotide differences (k), Tajima's D, Fu and Li's D, Fu and Li's F, Fu's Fs statistic based on ITS2 and *COI* sequences of *An. culicifacies*.**

| Region/Gene | Sample Size | S | h | $\pi$ | Hd | k | Tajima's D | Fu and Li's D | Fu and Li's F | Fu's Fs statistic |
|---|---|---|---|---|---|---|---|---|---|---|
| **ITS2** | 35 | 28 | 20 | 0.0252 | 0.661 | 8.5916 | 0.92220** | -1.24558** | -0.61994** | 5.755 |
| ***COI*** | 77 | 301 | 55 | 0.2752 | 0.981 | 104.0123 | 0.82540** | -2.48524* | -1.32233** | 5.036 |

* Significant P<0.05

**Not significant P>0.10

**Table 7. Pairwise fixation index (F$_{ST}$) values among populations based on ITS2 sequences of *An. culicifacies* species complex.**

|  | Iran | Cambodia | India | China | Sri Lanka |
|---|---|---|---|---|---|
| **Iran** | 0.00000 |  |  |  |  |
| **Cambodia** | 0.89474 | 0.00000 |  |  |  |
| **India** | 0.18429 | -0.57778 | 0.0000 |  |  |
| **China** | 0.92325 | 0.71429 | -0.04705 | 0.00000 |  |
| **Sri Lanka** | 0.89130 | 1.00000 | 0.55387 | 0.60000 | 0.00000 |

Matrix of significant F$_{ST}$ P values Significance Level = 0.0500

A relatively high haplotype diversity (Hd) values of 0.661 to 0.981 was observed in ITS2 and *COI* sequences respectively. The nucleotide diversity ($\pi$) and the average number of nucleotide differences (k) were higher in the analysis of *COI* sequences than in ITS2 sequences.

The G+C content was 0.591 at coding positions of 341.00 sites in ITS2 sequences and the total G+C content was 0.305 at coding positions of 378.00 sites in *COI* sequences. The total number of mutations was higher in *COI* sequences than in ITS2 sequences having 413 of a total number of mutations in *COI* and 28 total number of mutations in ITS2.

In both ITS2 and *COI* analysis statistically non-significant (P>0.10) positive values of Tajima's D values were obtained. There were negative values of Fu and Li's F and D and positive values of Fu's Fs statistic for both ITS2 and *COI* sequence analyses.

## Determination of genetic differentiation

**Genetic differentiation based on ITS2 sequences of *An. culicifacies* species complex.**
Pairwise fixation index (F$_{ST}$) values among populations of *An. culicifacies* species complex based on ITS2 sequences are shown in Table 7.

The commonly used values for F$_{ST}$ are as follows: 0 to 0.05, small; 0.05 to 0.15, moderate; 0.15 to 0.25, great; values above 0.25 indicate huge genetic differentiation [26].

According to Table 7, there were F$_{ST}$ values ranging from -0.57778 to 1.0000. The highest F$_{ST}$ value (1.0000) was obtained between Cambodia-Sri Lanka indicating huge genetic differentiation between populations. Negative F$_{ST}$ values which are effectively seen as zero were obtained for Cambodia-India, India-China indicating the absence of genetic subdivision between the considered populations.

**Genetic differentiation based on *COI* sequences of *An. culicifacies* species complex.**
Pairwise fixation index (F$_{ST}$) values among populations of *An. culicifacies* species complex based on *COI* sequences are shown in Table 8.

**Table 8. Pairwise fixation index (F$_{ST}$) values among populations based on *COI* sequences of *An. culicifacies* species complex.**

|  | Iran | Sri Lanka | Pakistan | Oman | India | Cambodia | Arab |
|---|---|---|---|---|---|---|---|
| **Iran** | 0.00000 |  |  |  |  |  |  |
| **Sri Lanka** | 0.58285 | 0.00000 |  |  |  |  |  |
| **Pakistan** | 0.35955 | 0.03758 | 0.00000 |  |  |  |  |
| **Oman** | -0.57294 | 0.68168 | 0.45902 | 0.00000 |  |  |  |
| **India** | 0.14620 | 0.20792 | 0.09479 | 0.09724 | 0.00000 |  |  |
| **Cambodia** | -0.70893 | 0.64652 | 0.42014 | 1.00000 | -0.10228 | 0.00000 |  |
| **Arab** | 0.23779 | -0.24728 | -0.67612 | 1.00000 | 0.05837 | 1.00000 | 0.00000 |

Matrix of significant F$_{ST}$ P values Significance Level = 0.0500

The commonly used values for $F_{ST}$ are as follows: 0 to 0.05, small; 0.05 to 0.15, moderate; 0.15 to 0.25, great; values above 0.25 indicate huge genetic differentiation [26].

According to Table 4, there were $F_{ST}$ values ranging from -0.70893 to 1.0000. The highest $F_{ST}$ value (1.0000) was obtained for Oman-Cambodia, Oman-Arab and Cambodia-Arab indicating huge genetic differentiation between populations. Negative $F_{ST}$ values which are effectively seen as zero were obtained for Iran-Oman, Iran-Cambodia, Sri Lanka-Arab, Pakistan-Arab and India-Cambodia indicating the absence of genetic subdivision between the considered populations when the allele frequencies of two subpopulations are very similar.

## Phylogeographic analysis of *An. culicifacies* species complex using ITS2 sequences

The phylogeographic tree of *An. culicifacies* species complex using ITS2 sequences is shown in Fig 5. According to the Fig 5, the diversification of the *Anopheles* genus was dated at 83.06 Mya (node A) with a 95% HPD ranging from 81.15 to 85.09 Mya in the Cretaceous. The diversification of *An. culicifacies* species complex was dated at 22.19 Mya (node B) with a 95% HPD ranging from 20.25 to 24.12 Mya in the late Miocene.

The species complex was split into two clades as in the phylogenetic analysis of ITS2 sequences. The diversification of clade 1 including sibling species A and D was dated at 16.25 Mya (node C) in Miocene. The sibling species A from Iran and both sibling species A and D

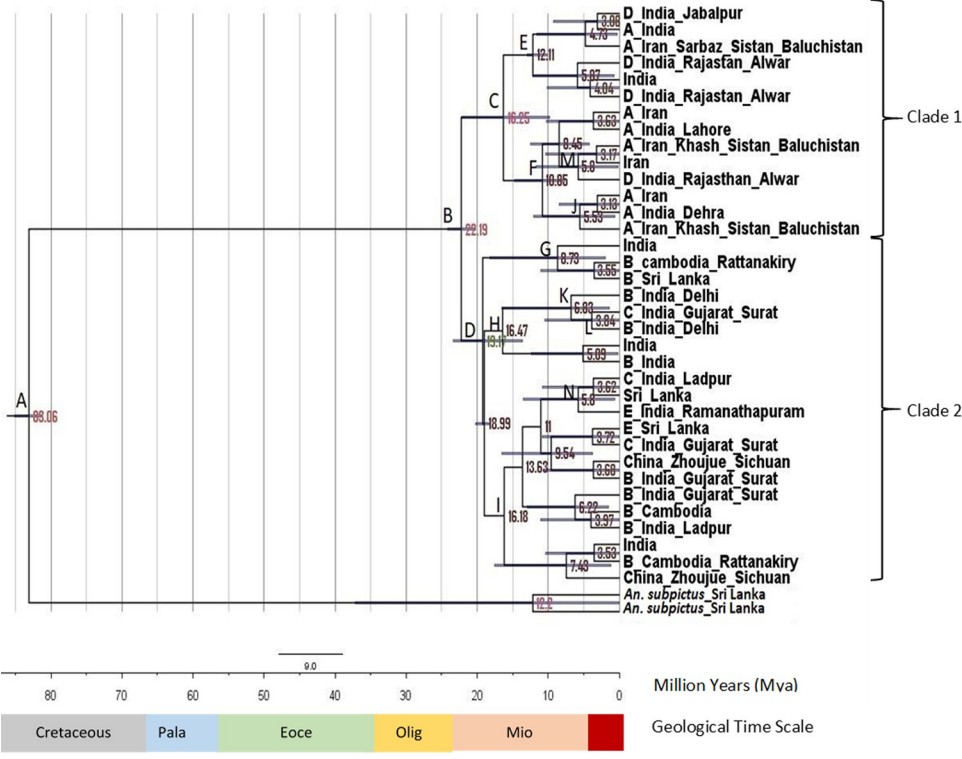

**Fig 5. Evolutionary timescale for *An. culicifacies* species complex generated by BEAST v1.8.2 software using ITS2 sequences.** 35 sequences, 435 characters. *An. subpictus* was used as the out-group. Numbers near the nodes designate the average divergence time estimated (Million Years, Mya). Geological time scale includes Cretaceous, Pala(eocene), Eoce(ne), Olig(ocene) and Mioc(ene) respectively. GenBank accession numbers of ITS2 sequences included in clades 1, 2 and outgroup were listed in S8 Table. Posterior probability values, Mean values of diversification times and 95% highest posterior density (HPD) of each nodes of the ITS2 phylogeographic tree is listed in S9 Table.

**Table 9. Diversification times of earliest members of the sibling species A, B, C, D and E of *An. culicifacies* species complex with countries based on the ITS2 phylogeographic tree.**

| Sibling species | Diversification (Mya) | Country |
|---|---|---|
| A | 5.53 (node J) | Iran |
| B | 6.83 (node K) | India |
| C | 3.84 (node L) | India |
| D | 5.80 (node M) | India |
| E | 5.80 (node N) | India |

from India were included in clade 1. There are two subclades inside clade 1 at node E and F that were dated at 12.11 Mya and 10.85 Mya separately. The diversification of clade 2 including sibling species B, C and E was 19.17 Mya (node D) in Miocene. The sibling species B from Sri Lanka, Cambodia and India, the sibling species C from India and the sibling species E from Sri Lanka were included in clade 2. There are three subclades inside clade 2 at nodes G, H and I that were dated at 8.73 Mya, 16.47 Mya and 16.18 Mya separately.

The diversification of the clade 1 including sibling species A and D was more recent than the clade 2 with sibling species B, C and E based on ITS2 phylogeographic analysis. The divergence of ITS2 sequences into clade 1 and clade 2 occurred prior to splitting into sibling species.

The diversification of the clade 1 including sibling species A and D was more recent than the clade 2 with sibling species B, C and E based on ITS2 phylogeographic analysis. The divergence of ITS2 sequences into clade 1 and clade 2 occurred prior to splitting into sibling species.

The diversification times of the earliest members of the sibling species A, B, C, D and E based on the ITS2 phylogeographic tree are listed in Table 9. According to that, the sibling species B can be considered as the earliest evolved sibling species and the sibling species C was the recently evolved sibling species based on ITS2 sequences.

## Phylogeographic tree of *An. culicifacies* species complex using *COI* sequences

The phylogeographic tree of *An. culicifacies* species complex using *COI* sequences was shown in Fig 6.

The split time of the most recent common ancestor of the *Anopheles* genus was obtained as 82.75 Mya in the Cretaceous (node A) with a 95% HPD (Height posterior density) ranging from 80.93 to 85.07 Mya. The split date of *An. culicifacies* species complex was 24.33 Mya shown in node B with a 95% HPD ranging from 22.37 to 26.22 Mya.

There were three clades diversified in the Miocene period. The diversification of clade 1 including sibling species from Iran, India, Pakistan, Oman and Sri Lanka was dated as 17.30 Mya (node C). The diversification of clade 2 and clade 3 was dated as 18.07 Mya (node D) and 17.92 Mya (node E) respectively. The sibling species from Sri Lanka, India and Pakistan were included in clade 2. The sibling species from Iran, India, Sri Lanka, Pakistan and the United Arab Emirates were included in clade 3.

There were three subclades identified within clade 1 arising at nodes F, G and H. The first subclade arising at node F was dated 10.11 Mya including sibling species A and D from India, and sibling species A from Pakistan, Iran and Oman. The second sub clade arising at node G was dated as 9.59 Mya including sibling species B and C from India, and sibling species B from Sri Lanka. The third sub clade arising at node H was dated 11.77 Mya including species B, C

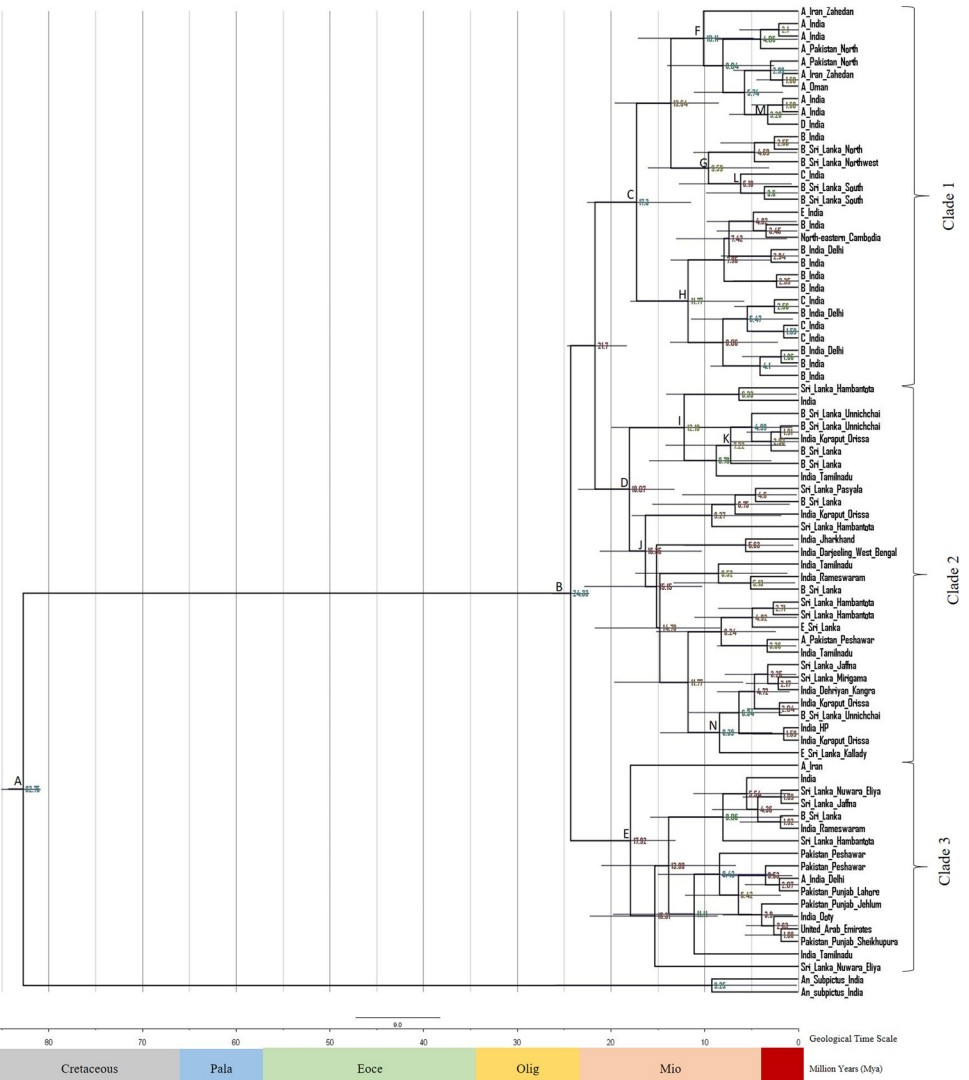

**Fig 6. Evolutionary timescale of *An. culicifacies* species complex generated by BEAST v1.8.2 software using *COI* sequences.** Sequences were retrieved from NCBI GenBank, 77 sequences, 450 characters. *An. subpictus* was used as the out-group. Numbers near the nodes designate the average divergence time estimated (Million Years, Mya). Geological time scale includes Cretaceous, Pala(eocene), Eoce(ne), Olig(ocene) and Mioc(ene) respectively. GenBank accession numbers of *COI* sequences included in clades 1 to 3 and outgroup were listed in S10 Table. Posterior probability values, Mean values of diversification times and 95% highest posterior density (HPD) of each nodes of the *COI* phylogeographic tree is listed in S11 Table.

and E from India and one specimen from Northeastern Cambodia. There were two sub clades identified within clade 2 arising at node I and J dated at 12.18 Mya and 16.35 Mya separately.

The diversification of sibling species of the species complex mainly occurred in Miocene. The diversification times of the earliest members of the sibling species A, B, C, D and E based on the *COI* phylogeographic tree were listed in Table 10. According to that, the sibling species A can be identified as the earliest evolved sibling species and the sibling species D was the recently evolved sibling species based on *COI* sequences.

**Table 10. Diversification times of earliest members of the sibling species A, B, C, D and E of *An. culicifacies* species complex with countries based on the *COI* phylogeographic tree.**

| Sibling species | Diversification (Mya) | Country |
|---|---|---|
| A | 17.92 (node E) | Iran |
| B | 7.22 (node K) | Sri Lanka |
| C | 6.18 (node L) | India |
| D | 3.28 (node M) | India |
| E | 8.39 (node N) | Sri Lanka |

## Discussion

Molecular taxonomy allows us to distinguish the mosquito vectors to species or sibling species level. Molecular systematics provides information on genetic diversity and supports predicting their evolution and phylogenetic relationships [31]. The analysis of phylogenetic and phylogeographic relationships, identification of haplotypes in different geographical areas, understanding of genetic diversity and genetic differentiation based on ITS2 and *COI* sequences of *An. culicifacies* species complex was carried out in this study. The Bayesian likelihood trees generated using ITS2 and *COI* sequences further supported the sibling species categorization and displayed the phylogenetic relationships between sibling species of *An. culicifacies* species complex with well-supported clades with high posterior probability values. In the phylogenetic analysis using ITS2 sequences, the generated Bayesian likelihood tree was dispersed into two main distinct clades including the sibling species B, C, and E in clade 1 and the sibling species A and D in clade 2. A considerable sequence divergence between these two clades and a strict sequence conservation within the members of a given clade was implied.

That result was the same as the results based on the primary and secondary structure of the ITS2 sequences. There are two clades of sibling species as A, D and B, C, and E of the species complex based on the primary and secondary structures of the ITS2 sequences [2].

There was more genetic change (or divergence) observed in the sibling species of A, and D than in the sibling species of B, C, and E after splitting from a common ancestor based on the branch length of the two clades of the ITS2 Bayesian likelihood tree. Another phylogenetic analysis of *An. culicifacies* species complex based on the ITS2 sequences performed using multiple aligned sequences with a neighbour-joining distance method, the species of A and D are under a less selective pressure and evolving faster compared to the sibling species in B, C and E. The members in a certain clade having a near sequence identity without any possibility of cross-breeding suggests that ITS2 regions are undergoing parallel evolution [8].

In the phylogenetic analysis using *COI* sequences, the generated Bayesian likelihood tree was dispersed into four clades producing a tree with a different topology than the ITS2 Bayesian likelihood tree. Only in the clades 2 and 3 of the *COI* Bayesian likelihood tree which includes the sibling species B, C, E in the clade 2 and sibling species A, D in the clade 3 were shown near relatedness to the clades of the ITS2 Bayesian likelihood tree. When considering the branch lengths of the clades 2 and 3, the obtained results were the same as the ITS2 phylogenetic analysis which was that sibling species A and D might have more genetic divergence than the sibling species B, C and E from their MRCA.

The generated *COI* Bayesian likelihood tree was divided into four clades as highly divergent *COI* sequences and the mitochondrial genome having a higher mutation rate than the nuclear genome. According to the generated two haplotype network, the total number of haplotypes was higher in the *COI* haplotype network than in the ITS2 haplotype network. Generally, the haploid mitochondrial genome has a higher mutation rate than the nuclear genome and the

occurring variations can produce different haplotypes. The considered number of *COI* sequences was also higher than the number of ITS2 sequences in this study.

The highest number of private haplotypes was recorded from India with high Hd values in both ITS2 and *COI* haplotype networks. A large effective population size in this area is represented by this private haplotype richness and the high haplotype diversity. The private haplotypes in a particular geographical area are indicators of a restricted gene flow from that area and a population that is adapted to local conditions [32].

There were lower Hd values obtained from Sri Lankan sequences among the sequences in both ITS2 and *COI* analysis as there are only two sibling species reported from Sri Lanka. In addition to that, the low haplotype diversity that occurred from bottlenecks might have resulted from the intense malaria control efforts, habitat fragmentation and disappearance of vertebrate blood meal sources [32]. It can become high in the future, through the imported cases resulting in a risk of disease re-emergence in Sri Lanka.

When considering the genetic diversity of *An. culicifacies* species complex based on the ITS2 and *COI* sequences recorded from all geographical regions, high Hd values were observed in both analyses, suggesting an extensive genetic diversity of the mosquitoes could be due to several facts such as including five sibling species in the complex, a large effective population size of mosquitoes, high gene flow among populations and favorable environmental conditions for mosquito breeding, regardless of control campaigns [33]. This high level of genetic diversity observed in this study implies that the possibility of *An. culicifacies* species complex maintaining a relatively high effective population size despite enormous population fluctuations.

A long-established mosquito population that has not faced recent bottlenecks and the possibility of the presence of genetically distant haplotypes in sympatry was indicated by the higher nucleotide diversity (π) and an average number of nucleotide differences (k) [33] in the analysis of *COI* sequences than the ITS2 sequences.

The null hypothesis of neutral evolution indicating that the mosquito populations in the study at mutation-drift equilibrium maintained a stable large population was accepted from non-significant Tajima's D values [33]. That was indicated by the non-significant Tajima's D values of both analyses of the ITS2 and *COI* sequences. There were positive values of Tajima's D in both ITS2 and *COI* analyses indicating a decreasing population size or balancing selection and a low level of both low and high-frequency polymorphisms [34].

There were negative values of Fu and Li's F and D in both analyses of the ITS2 and *COI* sequences from all countries in this study. An excess of singletons and a high number of unshared haplotypes were indicated by their negative values of it [35]. The deficiency of alleles from a recent population bottleneck can be due to a genetic drift causing reducing population size as indicated by positive values of Fu's Fs statistic [36].

Considering previous studies, the genetic diversity was well studied in *Ae. aegypti* using these neutrality test values. *Ae. aegypti* mosquitoes collected from different regions of Sri Lanka was genetically characterized in relation to *Ae. aegypti mosquitoes* found from various parts of the world based mitochondrial *COI* gene. A stable large population at mutation-drift equilibrium was suggested through the obtained statistically non-significant Tajima's D value as same as this study. Evolutionary fitness of the mosquito and stability of their populations is increased by the high genetic diversity and extensive genetic mixing [33].

According to the pairwise $F_{ST}$ values, a huge genetic differentiation between populations was observed from the highest $F_{ST}$ value of Cambodia-Sri Lanka in the ITS2 analysis and Oman-Cambodia, Oman-Arab and Cambodia-Arab in the *COI* analysis. A high genetic differentiation might be achieved due to the large geographical distance between the two locations [32]. When the distance between the populations increases, the genetic similarity between the populations was decreased. That suggests the isolation by distance that is derived from spatially

limited gene flow [37]. Negative $F_{ST}$ values which are effectively seen as zero were obtained for Cambodia-India, India-China in ITS2 analysis and Iran-Oman, Iran-Cambodia, Sri Lanka-Arab, Pakistan-Arab and India-Cambodia in *COI* analysis. $F_{ST}$ value is close to zero when the allele frequencies of two subpopulations are very similar and not differentiated [38]. High rates of gene flow and the absence of genetic subdivision between the considered populations were indicated by low $F_{ST}$ values.

*Anopheles* fossil record is very uncommon as there were only two *Anopheles* fossils recognized currently named *Anopheles* (*Nyssorhynchus*) *dominicanus* from the Late Eocene (33.9–41.3 Mya) being the oldest one and *Anopheles rottensis* from the Late Oligocene (13.8–33.9 Mya) being the most recent one [27]. The usage of those records seemed to be problematic based on the applied dating technique [39]. Therefore, the diversification date of the most recent common ancestor (MRCA) of the *Anopheles* genus was used to calibrate the phylogeographic trees in the study. The diversification of the *Anopheles* genus was dated in this study ranging from 81.15 to 85.09 Mya and 80.93 to 85.07 Mya based on ITS2 and *COI* phylogeographic analysis respectively in the Cretaceous.

The link between the diversification of anophelines and malaria parasites is still controversial [40]. The origin of *P. falciparum* was dated at 2.5 Mya and the initial radiation of mammalian *Plasmodium* at 12.8 Mya [41]. The divergence time of these species was younger than which inferred for their *Anopheles* vectors in our analysis.

The diversification of *An. culicifacies* species complex was obtained as ranging from 20.25 to 24.12 Mya and 22.37 to 26.22 Mya based on ITS2 and *COI* phylogeographic analysis respectively. That was nearly equal to the divergence of *An. culicifacies* based on mitochondrial genome PCG123 datasets were obtained as 21.76 Mya [42]. The diversification of the clade 2 including the sibling species B, C and E was earlier than the clade 1 with sibling species A and D as shown in the ITS2 phylogeographic analysis. The divergence of the clade 1 and 2 occurred prior to splitting into the sibling species. The diversification of sibling species of the species complex mainly occurred in the Miocene.

The split time of *An. subpictus* was more recent than the split time of *An. culicifacies* in this study that were obtained as 12.2 Mya and 9.25 Mya in ITS2 and *COI* phylogeographic analysis respectively. When considering other species of the genus *Anopheles*, the diversification of *Anopheles gambiae* / South East Asia-Oceania anophelines, *Anopheles darlingi* / *Anopheles albitarsis* complex and *Anopheles aquasalis* / *Anopheles albitarsis* complex were dated at 65.5 Mya, 38.98 Mya and 28.56 Mya respectively [29]. Therefore, the diversification of *An. culicifacies* species complex obtained in this study was more recent than these species of the *Anopheles* genus.

According to the obtained dates of the phylogeographic analysis, the sibling species B was considered as the earliest evolved sibling species based on ITS2 sequences and when considering *COI* sequences it was the sibling species A. The recently evolved sibling species were C and D respectively in ITS2 and *COI* analysis. Based on previous studies, it was mentioned that initially the sibling species A and B evolved from a common ancestor and the sibling species D diverged from A and the sibling species C from the sibling species B later [9]. The diversification times of the sibling species were not mentioned in previous studies.

The diversification dates of the earliest members of the sibling species are varied between ITS2 and *COI* analysis which can be due to the changes in the diversification of the nuclear genome and the mitochondrial genome. It is assumed that the mtDNA evolves at a faster rate than nuclear DNA (nuDNA) in animals and the ratio of mtDNA over nuDNA mutation rate varies between 2 and 6 in invertebrates including insects and arachnids [43].

Most of the intraspecific diversifications are seen in the Pliocene in the study. In ITS2 phylogeographic analysis, intraspecies diversification was observed mostly in the Pliocene period

in India, Iran, Cambodia, Sri Lanka and China as well as in *COI* analysis it was observed in countries of Pakistan, India, Sri Lanka and Cambodia. The climate is warm and humid during this period and tropical forest is likely to have been widespread across Southeast Asia and larval habitats abundant as well as the forest fragmentation is also increased. Therefore, it can be increased the niche availability and facilitate the spread [44]. All Pleistocene-dated divergence events can be observed in the *COI* phylogeographic analysis in the countries of Iran, India, Sri Lanka, Oman and the United Arab Emirates. It indicated that the Pleistocene glaciations have shaped genetic diversity within species and possibly to have retained sacks of forest habitat at intermediate elevations throughout the Pleistocene [44].

The possibility of *An. culicifacies* species complex maintaining a relatively high effective population size despite of the enormous population fluctuations was suggested. There are several limitations of this study as this was carried out using retrieved sequences of the NCBI database to fill the gaps of phylogeny, diversification and genetic diversity of *An. culicifacies* species complex. A concatenated phylogenetic tree could not be generated as both ITS2 and *COI* sequences were not reported from the same specimen. It is necessary to revisit the mosquito population to achieve success from the vector control strategies while incorporating the information on the genetic markers including both mitochondrial and nuclear markers to enhance the robustness of the reported observations.

## Conclusions

The importance of understanding the genetic diversity, phylogenetic and phylogeographic relationships of mosquito vectors using ITS2 and *COI* sequences of *An. culicifacies* species complex was focused. The sibling species of *An. culicifacies* species complex was broadly divided into two clades as A, D and B, C, E in both phylogenetic and phylogeographic analysis using ITS2 sequences. There were 3 to 4 clades in phylogenetic and phylogeographic analysis using *COI* sequences. There was more genetic divergence with less selective pressure and a recent diversification shown in the sibling species A and D than the sibling species B, C and E from their MRCA. The high genetic diversity of *An. culicifacies* species complex suggests the likelihood of maintaining a relatively high effective population size. The genetic diversities of vectors of mosquito-borne diseases should be considered in implementing effective vector control strategies as uniform control measures may not be equally effective for all the populations of a single species.

## Supporting information

**S1 Table. GenBank accession number, country of origin, sibling species of ITS2 sequences of *An. culicifacies*.**
(PDF)

**S2 Table. GenBank accession number, country of origin, sibling species of *COI* sequences of *An. culicifacies*.**
(PDF)

**S3 Table. GenBank accession numbers of ITS2 and *COI* sequences of the species used as outgroup of phylogenetic trees and phylogeographic trees.**
(PDF)

**S4 Table. GenBank accession numbers of the sequences included in clade 1, 2 and outgroup of Bayesian likelihood tree generated by MrBayes-3.2.5_WIN32_x86 software using ITS2 sequences of *An. culicifacies*.**
(PDF)

**S5 Table. GenBank accession numbers of the sequences included in clade 1 to 4 and outgroup of Bayesian likelihood tree generated by MrBayes-3.2.5_WIN32_x86 software using *COI* sequences of *An. culicifacies*.**
(PDF)

**S6 Table. Haplotype frequencies and haplotype diversities (Hd) in populations of ITS2 haplotype network of *An. culicifacies* species complex obtained using Arlequin version 3.5.2.2 and DnaSP v5 software.**
(PDF)

**S7 Table. Haplotype frequencies and haplotype diversities (Hd) in populations of *COI* haplotype network of *An. culicifacies* species complex obtained using Arlequin version 3.5.2.2 and DnaSP v5 software.**
(PDF)

**S8 Table. GenBank accession numbers of ITS2 sequences included in clade 1, 2 and outgroup of phylogeographic tree generated by BEAST v1.8.2 software using ITS2 sequences of *An. culicifacies*.**
(PDF)

**S9 Table. Posterior probability values, Mean values of diversification times and 95% highest posterior density (HPD) of each nodes of the ITS2 phylogeographic tree.**
(PDF)

**S10 Table. GenBank accession numbers of *COI* sequences included in clade 1 to 3 and outgroup of phylogeographic tree generated by BEAST v1.8.2 software using *COI* sequences of *An. culicifacies*.**
(PDF)

**S11 Table. Posterior probability values, Mean values of diversification times and 95% highest posterior density (HPD) of each nodes of the *COI* phylogeographic tree.**
(PDF)

## Acknowledgments

All the members of the Centre for Biotechnology and Genetics and Molecular Biology Unit at the University of Sri Jayewardenepura, Sri Lanka were greatly acknowledged for the assistance rendered during this study.

## Author Contributions

**Conceptualization:** B. G. D. N. K. De Silva.

**Data curation:** R. A. S. Rathnayake.

**Funding acquisition:** B. G. D. N. K. De Silva.

**Methodology:** R. A. S. Rathnayake, W. M. M. Wedage.

**Software:** R. A. S. Rathnayake, W. M. M. Wedage, L. S. Muthukumarana.

**Supervision:** B. G. D. N. K. De Silva.

**Writing – original draft:** R. A. S. Rathnayake.

**Writing – review & editing:** B. G. D. N. K. De Silva.

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
