## [Decision Letter · Decision Letter 0]

21 May 2023

PONE-D-23-08529Genetic diversity, phylogenetic and phylogeographic analysis of Anopheles culicifacies species complex using ITS2 and COI sequencesPLOS ONE

Dear Dr. De Silva,

Thank you for submitting your manuscript to PLOS ONE. After careful consideration, we feel that it has merit but does not fully meet PLOS ONE’s publication criteria as it currently stands. Therefore, we invite you to submit a revised version of the manuscript that addresses the points raised during the review process.

We look forward to receiving your revised manuscript.

Kind regards,

Maria Stefania Latrofa

Academic Editor

PLOS ONE

Journal Requirements: 

3. Thank you for stating the following financial disclosure: "The author(s) received no specific funding for this work."

4. Please remove your figures from within your manuscript file, leaving only the individual TIFF/EPS image files, uploaded separately. These will be automatically included in the reviewers’ PDF.

5. Please ensure that you refer to Figures 8 and 9 in your text as, if accepted, production will need this reference to link the reader to the figure.

6. We note that Figures 1, 4 and 6 in your submission contain map images which may be copyrighted. All PLOS content is published under the Creative Commons Attribution License (CC BY 4.0), which means that the manuscript, images, and Supporting Information files will be freely available online, and any third party is permitted to access, download, copy, distribute, and use these materials in any way, even commercially, with proper attribution. For these reasons, we cannot publish previously copyrighted maps or satellite images created using proprietary data, such as Google software (Google Maps, Street View, and Earth). For more information, see our copyright guidelines: http://journals.plos.org/plosone/s/licenses-and-copyright.

(1) You may seek permission from the original copyright holder of Figures 1, 4 and 6 to publish the content specifically under the CC BY 4.0 license.  

Reviewers' comments:

Reviewer's Responses to Questions

**Comments to the Author**

1. Is the manuscript technically sound, and do the data support the conclusions?

Reviewer #1: Partly

Reviewer #2: Yes

2. Has the statistical analysis been performed appropriately and rigorously? 

Reviewer #1: Yes

Reviewer #2: Yes

3. Have the authors made all data underlying the findings in their manuscript fully available?

Reviewer #1: Yes

Reviewer #2: Yes

4. Is the manuscript presented in an intelligible fashion and written in standard English?

Reviewer #1: No

Reviewer #2: Yes

5. Review Comments to the Author

Reviewer #1: I have reviewed the manuscript titled “Genetic diversity, phylogenetic and phylogeographic analysis of Anopheles culicifacies species complex using ITS2 and COI sequences” and find the work to be interesting and potentially valuable to the scientific community. However, I have some major concerns that need to be addressed before publishing.

Overall, the manuscript presents a clear research question. The methodology is fairly sound, and the results are generally well-supported. However, there are several major issues that must be addressed before it can be published.

One of my main concerns is the lack of analysis based on the sibling species. The genetic diversity values focus on An. culicifacies as a complex rather than the sibling species. For example, when presenting data on the haplotypes, rather than stating the haplotype numbers/diversity for the entire complex better to give details on individual sibling species. Also, I suggest authors to do a population structure analysis to complete the study. I would like to suggest the authors to revise the abstract and the conclusion.

The writing style of the manuscript needs significant improvement. The language is often unclear and confusing, making it difficult for reader to understand the meaning. The manuscript needs to be revised for clarity and coherence. Also, the table and figure titles are not up to the journal standards.

In light of these concerns, I recommend that the manuscript to be resubmitted after making the suggested major revision. Therefore, I would like to suggest the authors to be given the opportunity to revise and resubmit their work. I believe that with significant revisions, this manuscript has the potential to make a valuable contribution to the scientific literature.

Refer the attached PDF with detailed comments and suggestions.

Abstract

I would like to request that you revise your abstract to better reflect your methods and results. While your current abstract provides a general overview of your study, it would be beneficial if you include more specific details regarding your methods and results. This will help better understand the scope of your study and the significance your findings.

For example, in your methods section, you may want to include the number of COI and ITS2 sequences used, the countries covered etc. In the results section information such as details on haplotypes and genetic diversity information of each sibling species would have strengthen the abstract.

I understand that abstracts are often limited in length, but I believe that providing more details information will greatly enhance the quality and impact of your manuscript.

Introduction

The authors could have provided adequate information on the selection of the two genetic markers COI and ITS2 for the study referring to previous studies. Also, the authors have not provided sufficient background information to reflect the current state of research in the field.

Including these details in the introduction, will be able to quickly understand the key aspects of your study.

Methods

This section can be improved by giving details on the number of sequences extracted from each sibling species. Further, the number of sequences from each country also could have been presented as a percentage which will reflect the sequence representation from each country.

Importantly I wonder why the authors have used DnaSP software to analyze pairwise differences. Arlequin is a more sensitive software to do the analysis. I suggest authors to recalculate Fst values using Arlequin.

Further, very basic information of the analysis has been included in the text which can be presented in more advance form to match the journal standards.

Results

I would like to suggest the authors to present results based on the sibling species too, along with the results for overall complex.

Discussion

The discussion section is also insufficient. The authors have not provided a thorough interpretation of their results, not have they fully discussed the implications of their findings. Additionally, the limitations of the study have not fully addressed.

Conclusion

Can be improved by summarizing the facts and presenting only the main outcomes of the study.

Reviewer #2: The manuscript PONE-D-23-08529, entitled “Genetic diversity, phylogenetic and phylogeographic analysis of Anopheles culicifacies species complex using ITS2 and COI sequences” is a molecular data analyses to infer the phylogenetic and phylogeographic relationships of An. culicifacies species complex targeting ITS2 and COI sequences available in the GenBank.

Here in this article, authors are assessing the phylogenetic and phylogeographic relationships between the sibling species of the An. culicifacies species complex while observing their genetic diversity and genetic differentiation. In general, the article is well written except for some repetition of results in the discussion section. I have minor suggestions, which have been mentioned in the attached pdf.

6. PLOS authors have the option to publish the peer review history of their article (what does this mean?). If published, this will include your full peer review and any attached files.

Reviewer #1: No

Reviewer #2: No

---

## [Author Response · Author response to Decision Letter 0]

6 Jul 2023

Reviewer #1

We thank reviewer 1 for his/her helpful comments on our manuscript. Below changes were made accordingly in the manuscript.

Abstract

Comment: Line 11: Would have presented result section more logically. Even the methodology can be improved and presented with more clarity.

Response: Results and Methodology section were modified according to the comments of each section. 

Comment: Line 23: Word ‘many’ Need to give the exact number.

Response: The exact number of clades of the phylogenetic and phylogeographic trees of COI was mentioned.

(Line numbers: 23-24)

Comment: I would like to request that you revise your abstract to better reflect your methods and results. While your current abstract provides a general overview of your study, it would be beneficial if you include more specific details regarding your methods and results. This will help better understand the scope of your study and the significance your findings.

For example, in your methods section, you may want to include the number of COI and ITS2 sequences used, the countries covered etc. In the results section information such as details on haplotypes and genetic diversity information of each sibling species would have strengthen the abstract.

I understand that abstracts are often limited in length, but I believe that providing more details information will greatly enhance the quality and impact of your manuscript.

Response: Abstract was modified adding the number of sequences of ITS2 and COI and reported countries used for the analysis. More modifications were unable to add as the length of the abstract section is limited.

(Line numbers: 16-19). 

Comment : Line 28: Bottlenecks or due to presence of only two sibling species.

Response : According to the 33rd reference, the low haplotype can be obtained from the bottlenecks resulted from the intense control efforts, habitat fragmentation and disappearance of vertebrate blood meal sources etc. 

Mayoke A, Muya SM, Bateta R, Mireji PO, Okoth SO, Onyoyo SG, et al. Genetic diversity and phylogenetic relationships of tsetse flies of the palpalis group in Congo Brazzaville based on mitochondrial cox1 gene sequences. Parasites and Vectors. 2020; 13(1): 1–16. doi: 10.1186/s13071-020-04120-3.

(Line number: 709)

Comment: Line 30: It’s obviously to have high genetic diversity as there are five sibling species in the complex.

Response :The changes are done accordingly.

(Line numbers: 31-33)

Introduction

Comment : Line 73, 74: I would like to suggest the author to include a para or few sentences on the use of ITS2 and COI in analyzing genetic diversity of An. culicifacies or any Anopheline species in the world. 

The authors could have provided adequate information on the selection of the two genetic markers COI and ITS2 for the study referring to previous studies. Also, the authors have not provided sufficient background information to reflect the current state of research in the field.

Including these details in the introduction, will be able to quickly understand the key aspects of your study.

Response :A paragraph was added about the two genetic markers of ITS2 and COI based on previous studies. Changes were made accordingly to the comments.(Line numbers: 73-88)

Comment : COI can be in italics as it is a gene.

Response: We have kept COI earlier as non-italicized form according to some research papers published. 

Corrected in the manuscript according to reviewers’ comments. 

Materials and methods

Comment : This section can be improved by giving details on the number of sequences extracted from each sibling species. Further, the number of sequences from each country also could have been presented as a percentage which will reflect the sequence representation from each country.

Response: Changes were made accordingly. Table 1 and 2 were added for giving details on the number of sequences extracted from each sibling species and the number of sequences from each country.

(Line number: 104 Table 1)

(Line number: 106 Table 2)

Comment : Importantly I wonder why the authors have used DnaSP software to analyze pairwise differences. Arlequin is a more sensitive software to do the analysis. I suggest authors to recalculate Fst values using Arlequin.

Response: Pairwise differences were recalculated using Arlequin. Table 7 and 8 were changed 

(Line number: 284 Table 7) 

(Line number: 298 Table 8)

Comment : Further, very basic information of the analysis has been included in the text which can be presented in more advance form to match the journal standards.

Response: Basic information was changed according to the given comments.

Comment : Line 76: Better to mention the countries that you could not find any sequence records although the mosquito is present.

Response: Unable to find the countries without any sequence records although the mosquito is present.

Comment : Line 78: The words "Phylogenetic analysis"are repeating withing the sentence and would suggest to rewrite the sentence.

Response :The sentence was rewritten as ‘Four sequences of Aedes aegypti and two sequences of An. subpictus were used as the outgroup of the phylogenetic and phylogeographic analyses respectively’.

(Line numbers: 96-98)

Comment : Line 80-81: Repetition of the sentence 1.

Response :The sentence was removed as the repetition.

Comment : Line 83: Obviously you cannot use these. No need to present such basic information.

Response :The sentence was removed.

Comment : Line 85-86: Sri Lanka, India, Pakistan, Iran and Saudi Arabia are not south east Asian countries. This must be map of Asia not south east Asia. This needs to be corrected throughout the manuscript.

Response : It was corrected as ‘Asia’. The map (Fig 1)was removed as it was unable to obtain permission from the original copyright holder to publish under the CC BY 4.0 license. Figure numbers of the rest were changed accordingly.

(Line numbers: 100-101)

Comment : Line 89: As these are basic steps in phylogenetic analysis not necessary to mention this.

Response : Change the sentence accordingly.

(Line numbers: 109-110)

Comment : Line 103-104: suggest to delete as these are basic steps done during the analysis.

Response : The basic steps were removed from the sentence.

Comment : Line 107: Any special reason for using Arlequin software to determine the haplotype frequency rather than the Dnasp software?

Response: Because it is more sensitive than using DnaSP software as it allows to carry out tests of population genetics with a large set of basic methods and statistical tests. These statistical tests are powerful and minimize hidden assumptions.

Comment : Line 118-119: The FST values are best calculated by the Arlequin software. Why Dnasp? Arlequin is more sensitive in calculating the Fst values.

Response : FST values were recalculated using Arlequin software and changes were made accordingly in sections of materials and methods 

(Line numbers: 133-135)

and results.

(Line number: 284 Table 7) 

(Line number: 298 Table 8)

Results

Comment : Suggest to mention the number of sequences from each country for each molecular marker as a percentage. Because if the number of sequences reported from one country is lower obviously it will give lower haplotype numbers and diversities.

Response : Table 4 and 5 was added giving details on the sibling species belonging to each haplogroup and reported countries.

(Line number: 220 Table 4)

(Line number: 256 Table 5)

Comment : I would like to suggest the authors to present results based on the sibling species too, along with the results for overall complex.

Response : The results were present based on the sibling species. 

(Line numbers: 220 and 256)

Comment : Line 134-142: These are not your results. These are data that you have retrieved from the Genbank. What is available in the Genbank.

Response : The ITS2 sequences were retrieved from the NCBI public database. The phylogenetic tree was generated using those sequences in this analysis as there were no previous studies on generating a phylogenetic tree of An. culicifacies using ITS2 sequences reported from different countries such as Sri Lanka, India, Pakistan, Oman, Iran, Cambodia, Arab and China.

Comment : Line 154: Better to label nodes using letters other than the names of sibling species.

Response : The nodes were labelled using letters P onwards.

Comment : Line 159: Correct grammer.

Response : The sentence was corrected accordingly. 

(Line numbers: 167-168)

Comment : Line 199: Since the authors talk about five different sibling species, I would like to recommend the authors to present numbers of haplotypes and their distribution based on the sibling species.

Response : Table 4 and 5 was added giving details on the sibling species belonging to each haplogroup and reported countries.

(Line number: 220 Table 4)

(Line number: 256 Table 5)

Comment : Line 216: This map (Fig 4) does not show any information on the haplotypes. Suggest to remove.

Response : The Fig 4 was added to show the countries of each haplotype of ITS2 belongs. Fig 4 was removed from the manuscript based on the reviewer’s comment. Figure numbers of the rest were changed accordingly.

Comment : Line 226: As the target of the study is to determine the haplotype distribution of sibling species, would like to suggest mentioning the sibling species coming under each haplogroup.

Response : Table 4 and 5 was added giving details on the sibling species belongs to each haplogroup and reported countries.

(Line number: 220 Table 4)

(Line number: 256 Table 5)

Comment : Line 228: H_2 and H_11 Can be included within brackets.

Response : Brackets were added for H_2 and H_11.

(Line number: 211)

Comment : Line 228: Word ‘base’. This word does not match here.

Response : Change the sentence accordingly.

(Line numbers: 210-211)

Comment : Line 239-240: ‘based on the United Arab Emirates 77 COI sequences used in the analysis’ highlighted.

Response : The sentence was corrected.

(Line number: 223-225)

Comment : Line 249: Again I would like to suggest you to mention the sibling species coming under each haplogroup in addition to the countries.

Response : Table 4 and 5 was added giving details on the sibling species belongs to each haplogroup and reported countries.

(Line number: 220 Table 4)

(Line number: 256 Table 5)

Comment : Line 267: Fig 6 caption was highlighted.

Response : The Fig 6 was added to show the countries of each haplotype of COI belongs. Fig 6 was removed from the manuscript based on the reviewer’s comment. Figure numbers of the rest were changed accordingly.

Comment : Line 295: Can you really say this value as high? Better to say relatively high.

Response : The value was interpreted as relatively high.

(Line number: 269)

Comment : Line 306: Arlequin is more sensitive in determining FST values.

Response : Table 7 and Table 8 were changed according to the reanalyzed FST values using Arlequin software. The manuscript was changed according to the newly obtained FST values.

(Line number: 284 Table 7) 

(Line number :298 Table 8)

Discussion

Comment : Line 463: Since you are not discussing about genetic diversity of individual sibling species it’s obvious to get higher genetic diversities as there are five sibling species representing the complex. Therefore, it’s better to present the diversities of each sibling species too.

Response : This study is aimed to study about the sibling species complex. The sequences used here are retrieved sequences from the NCBI data base. There are 8 and 36 sequences from each ITS2 and COI which are not categorized into any sibling species level. Therefore, the genetic diversity was obtained for all ITS2 and COI sequences separately.

Comment : When you are presenting the diversity of the species complex would have compared with at least other mosquito species complexes.

Response : A new paragraph was added about the genetic diversity of other mosquito species.

(Line number: 467-473)

Comment : Line 469-470: Several sibling species has been clustered together in the phylogenetic trees with both markers. Then how can you say that both markers supported sibling species identification?

Response : The sentence was corrected accordingly.

(Line number: 390-393)

Comment : Line 477-478: The sentence is not clear. This is comparison with a previous study?

Response : The sentence shows a comparison with a previous study based on 2nd reference.

(Line numbers: 398-400)

Shanika KHS, Harischandra IN, De Silva BGDNK. Nucleotide sequence and secondary structure variations in ITS2-rDNA region of the members of Anopheles culicifacies (Diptera : Culicidae) species complex Nucleotide Sequence and Secondary Structure Variations in ITS2-rDNA Region of the Members of Anopheles culicifacies (Diptera: Culicidae) Species Complex. Vidyodaya J. of Sci. 2016; 20: 26-40

(Line number: 602)

Comment : Line 481-485: Not clear.

Response : This was added to show the methods that are used to differentiate five sibling species separately. Although, there were two clades in ITS2 phylogenetic tree with the sibling species A,D and B,C,E, all of these five sibling species can be differentiated from each other based on mentioned methods. That paragraph was removed in the manuscript as it is confusing.

(Line numbers: 401-408)

Comment : Line 496-499: Not clear. It’s not clear how you are going to justify or relate your results with these results

Response : The sentence was corrected accordingly.

(Line numbers: 416-421)

Comment : Line 505-506: Incomplete sentence.

Response : The sentence was corrected.

(Line number: 426-428)

Comment : Line 508-510: As you have not used similar number of sequences of both markers in the analysis (as you have mentioned below), you cannot give this statement. You just can present this as a common statement about these markers.

Response : The sentence was corrected accordingly.

(Line number: 429-431)

Comment : Line 517-518: Sri Lanka has reported only two sibling species so far. Therefore, isn't it the reason for this low Hd?

Response : Yes. It can be one reason and in addition to that intense malaria control efforts also can influence this. The sentence was changed accordingly.

(Line numbers: 439-443)

Comment : Line 520-521: The meaning is not clear.

Response : Hd values can become high in the future, through imported cases resulting in a risk of disease re-emergence.

(Line numbers: 442-443)

Comment : Line 523: High Hd values - This is because there are five sibling species coming under the complex.

Response : The sentence was changed accordingly.

(Line number: 447)

Comment : Line 555: Although India and Pakistan are connected via land and closer to each other the Fst values were higher than the value between Sri Lankan and Pakistan which are separated by both land and sea. How can you justify this?

Response : The FST was recalculated using Arlequin software as it is more sensitive. The previous results were changed based on the new values.

(Line numbers: 474-485)

Comment : Line 587-589: Why do you think the reason for this difference?

Response : In previous studies, based on reference 9, the method used for obtaining the diversification dates was not mentioned. The difference in these results can be due to the sequence differences of ITS2 and COI.

(Line numbers: 518-520)

Surendran SN, Ramasamy R. The Anopheles culicifacies and An. subpictus species complexes in Sri Lanka and their implications for Malaria control in the country. Trop. Med. Health. 2010; 38(1): 1–11. doi: 10.2149/tmh.2009-12.

(Line number: 628)

Comment : Line 596: Incomplete sentence.

Response : The sentence was corrected.

(Line number: 527)

Comment : The discussion section is also insufficient. The authors have not provided a thorough interpretation of their results, not have they fully discussed the implications of their findings. Additionally, the limitations of the study have not fully addressed.

Response : Corrected the discussion section according to the given comments and the limitations were discussed.

(Line numbers: 539-543)

Conclusion

Comment : Line 617-619: Can summarize what you are trying to say from this sentence.

Can be improved by summarizing the facts and presenting only the main outcomes of the study.

Response : The changes were done accordingly.

(Line numbers: 552-554)

Editorial and Data Presentation Modifications?

Comment : I have reviewed the manuscript titled “Genetic diversity, phylogenetic and phylogeographic analysis of Anopheles culicifacies species complex using ITS2 and COI sequences” and find the work to be interesting and potentially valuable to the scientific community. However, I have some major concerns that need to be addressed before publishing.

Response : Corrected accordingly.

Summary and General Comments

Comment : Overall, the manuscript presents a clear research question. The methodology is fairly sound, and the results are generally well-supported. However, there are several major issues that must be addressed before it can be published.

One of my main concerns is the lack of analysis based on the sibling species. The genetic diversity values focus on An. culicifacies as a complex rather than the sibling species. For example, when presenting data on the haplotypes, rather than stating the haplotype numbers/diversity for the entire complex better to give details on individual sibling species. Also, I suggest authors to do a population structure analysis to complete the study. I would like to suggest the authors to revise the abstract and the conclusion.

The writing style of the manuscript needs significant improvement. The language is often unclear and confusing, making it difficult for reader to understand the meaning. The manuscript needs to be revised for clarity and coherence. Also, the table and figure titles are not up to the journal standards.

Response: Manuscript was corrected accordingly.

The population structure analysis was not able to be proceeded as the lack of sufficient data such as GPS points of the sequences. The objectives of this study were mainly focused on genetic diversity, phylogeny and diversification of An. culicifacies species complex.

Reviewer #2

We thank reviewer 2 for his/her helpful comments on our manuscript. Below changes were made accordingly in the manuscript.

Introduction

Comment : Line 35: Space between two sentences.

Response : Corrected accordingly.

(Line number: 36)

Comment : Line 37: An. culicifacies,Write full name at the starting of the sentence.

Response : Corrected as Anopheles culicifacies.

(Line number: 38)

Comment : Line 38-40: Check the sentence construction.

Response : Corrected the sentence accordingly.

(Line numbers: 39-41)

Comment : Line 43,45: Both reference in same bracket.

Response : Multiple references were added in a single bracket throughout the manuscript.

(Line numbers: 41, 44, 47, 52)

Comment : Line 47: Comma after Species.

Response : It was corrected.

(Line number: 49)

Comment : Line 63: In-depth instead ‘A depth’.

Response : The sentence was corrected accordingly.

(Line number: 65)

Comment : Line 71: Word ‘Hence’ added.

Response : The sentence was corrected accordingly.

(Line number: 89)

Materials and Methods

Comment : Line 79: Mentioning first time. So write in full.

Response : Corrected as Anopheles subpictus.

(Line number 97)

Comment : Line 83: please check it.

Response : The count of ITS2 and COI sequences was rechecked in the NCBI Public database.

(Line numbers: 98-99)

Comment : Line 98: Word ‘Networks’ was corrected as network.

Response : It was corrected as the network.

(Line number: 120)

Results

Comment : Line134-142: ITS2 phylogenetic tree: Scientific name in italics.

Response : Scientific names were corrected in italics in phylogenetic trees and phylogeographic trees.

(Line number: 151 Fig 1)

(Line number: 179 Fig 2)

(Line number: 318 Fig 5)

(Line number: 352 Fig 6)

Comment : Line 151,152: Please rewrite this sentence.

Response : The sentence was changed accordingly.

(Line numbers: 159-160)

Discussion

Comment : Line 485: I think this way it will be better. "species A-B and A-C".

Response : That paragraph was removed as it is confusing.

(Line numbers: 401-408)

Comment : Line 486: between species A and B.

Response : That paragraph was removed as it is confusing.

(Line numbers: 401-408)

Comment : Line 487: species A-B and B-C.

Response : That paragraph was removed as it is confusing.

(Line numbers: 401-408)

References

Comment : Line 717: Reference number 19: Italic-Phlebotomus papatasi.

Response : It was italicized in Reference 25. Reference numbers were changed according to the corrections.

(Line number: 682)

Comment : Line 726: Reference number 21: Italic-Sporothrix schenckii.

Response : It was italicized in Reference 27.

Reference numbers were changed according to the corrections.

(Line number: 691)

Comment : Line 734: Reference number 24: Italic- Anopheles aquasalis.

Response : It was italicized in Reference 30.

Reference numbers were changed according to the corrections.

(Line number: 699)

Editorial and Data Presentation Modifications?

Comment : The manuscript PONE-D-23-08529, entitled “Genetic diversity, phylogenetic and phylogeographic analysis of Anopheles culicifacies species complex using ITS2 and COI sequences” is a molecular data analyses to infer the phylogenetic and phylogeographic relationships of An. culicifacies species complex targeting ITS2 and COI sequences available in the GenBank.

Here in this article, authors are assessing the phylogenetic and phylogeographic relationships between the sibling species of the An. culicifacies species complex while observing their genetic diversity and genetic differentiation. In general, the article is well written except for some repetition of results in the discussion section. I have minor suggestions, which have been mentioned in the attached pdf.

Response : Corrected accordingly.

Editorial and Data Presentation Modifications?

Comment : The present study needs minor revisions. Details are given in different sections.

Response : Corrected accordingly.

---

## [Decision Letter · Decision Letter 1]

3 Aug 2023

Genetic diversity, phylogenetic and phylogeographic analysis of Anopheles culicifacies species complex using ITS2 and COI sequences

PONE-D-23-08529R1

Dear Dr. De Silva,

We’re pleased to inform you that your manuscript has been judged scientifically suitable for publication and will be formally accepted for publication once it meets all outstanding technical requirements.

Kind regards,

Maria Stefania Latrofa

Academic Editor

PLOS ONE

Reviewer's Responses to Questions

**Comments to the Author**

1. If the authors have adequately addressed your comments raised in a previous round of review and you feel that this manuscript is now acceptable for publication, you may indicate that here to bypass the “Comments to the Author” section, enter your conflict of interest statement in the “Confidential to Editor” section, and submit your "Accept" recommendation.

Reviewer #2: All comments have been addressed

2. Is the manuscript technically sound, and do the data support the conclusions?

Reviewer #2: Yes

3. Has the statistical analysis been performed appropriately and rigorously? 

Reviewer #2: N/A

4. Have the authors made all data underlying the findings in their manuscript fully available?

Reviewer #2: Yes

5. Is the manuscript presented in an intelligible fashion and written in standard English?

Reviewer #2: Yes

6. Review Comments to the Author

Reviewer #2: I am pleased to inform that the R1 has considerably improved the paper. Authors have carefully addressed the reviewer’s comments. Hence in my opinion this paper can be accepted for publication in PLOS ONE.

7. PLOS authors have the option to publish the peer review history of their article (what does this mean?). If published, this will include your full peer review and any attached files.

Reviewer #2: No

---

## [Editor Report · Acceptance letter]

7 Aug 2023

PONE-D-23-08529R1 

Genetic diversity, phylogenetic and phylogeographic analysis of *Anopheles culicifacies* species complex using ITS2 and *COI* sequences 

Dear Dr. De Silva:

I'm pleased to inform you that your manuscript has been deemed suitable for publication in PLOS ONE. Congratulations! Your manuscript is now with our production department. 

Kind regards, 

on behalf of

Dr. Maria Stefania Latrofa 

Academic Editor

PLOS ONE